# Expression Profiling of Extracellular Matrix Genes Reveals Global and Entity-Specific Characteristics in Adenoid Cystic, Mucoepidermoid and Salivary Duct Carcinomas

**DOI:** 10.3390/cancers12092466

**Published:** 2020-08-31

**Authors:** Christoph Arolt, Moritz Meyer, Franziska Hoffmann, Svenja Wagener-Ryczek, David Schwarz, Lisa Nachtsheim, Dirk Beutner, Margarete Odenthal, Orlando Guntinas-Lichius, Reinhard Buettner, Ferdinand von Eggeling, Jens Peter Klußmann, Alexander Quaas

**Affiliations:** 1Institute of Pathology, Medical Faculty, University of Cologne, 50937 Cologne, Germany; svenja.wagener@uk-koeln.de (S.W.-R.); m.odenthal@uni-koeln.de (M.O.); reinhard.buettner@uk-koeln.de (R.B.); alexander.quaas@uk-koeln.de (A.Q.); 2Department of Otorhinolaryngology, Head and Neck Surgery, Medical Faculty, University of Cologne, 50937 Cologne, Germany; Moritz.Meyer@uk-essen.de (M.M.); david.schwarz@uk-koeln.de (D.S.); lisa.nachtsheim@uk-koeln.de (L.N.); jens.klussmann@uk-koeln.de (J.P.K.); 3Department of Otorhinolaryngology, Head and Neck Surgery, University Hospital Essen, University Duisburg-Essen, 45147 Essen, Germany; 4Department of Otorhinolaryngology, MALDI Imaging and Innovative Biophotonics, Jena University Hospital, 07747 Jena, Germany; Franziska.Hoffmann@med.uni-jena.de; 5Department of Otorhinolaryngology, Head and Neck Surgery, University Medical Center Göttingen, 37075 Göttingen, Germany; dirk.beutner@med.uni-goettingen.de; 6Department of Otorhinolaryngology, Head and Neck Surgery, Jena University Hospital, 07747 Jena, Germany; Orlando.Guntinas@med.uni-jena.de; 7Department of Otorhinolaryngology, MALDI Imaging, Core Unit Proteome Analysis, DFG Core Unit Jena Biophotonic and Imaging Laboratory (JBIL), Jena University Hospital, 07747 Jena, Germany; Ferdinand.von_Eggeling@med.uni-jena.de

**Keywords:** salivary gland carcinoma, extracellular matrix (ECM), *COL27A1*, COL11A1

## Abstract

**Simple Summary:**

The extracellular matrix (ECM), an important factor in tumour metastasis and therapy resistance, has not been studied in salivary gland carcinomas (SGC), so far. In this retrospective study, we profiled the RNA expression of 28 ECM-related genes in 11 adenoid cystic (AdCy), 14 mucoepidermoid (MuEp) and 9 salivary duct carcinomas (SaDu). Also, we validated our results in a multimodal approach. MuEp and SaDu shared a common gene signature involving an overexpression of *COL11A1*. In contrast, nonhierarchical clustering revealed a more specific gene expression pattern for AdCy, characterized by overexpression of *COL27A1*. In situ studies at RNA level indicated that in AdCy, ECM production results from tumour cells and not from cancer-associated fibroblasts as is the case in MuEp and SaDu. For the first time, we characterized the ECM composition in SGC and identified several differentially expressed genes, which are potential therapeutic targets.

**Abstract:**

The composition of the extracellular matrix (ECM) plays a pivotal role in tumour initiation, metastasis and therapy resistance. Until now, the ECM composition of salivary gland carcinomas (SGC) has not been studied. We quantitatively analysed the mRNA of 28 ECM-related genes of 34 adenoid cystic (AdCy; *n* = 11), mucoepidermoid (MuEp; *n* = 14) and salivary duct carcinomas (SaDu; *n* = 9). An incremental overexpression of six collagens (including *COL11A1*) and four glycoproteins from MuEp and SaDu suggested a common ECM alteration. Conversely, AdCy and MuEp displayed a distinct overexpression of *COL27A1* and *LAMB3*, respectively. Nonhierarchical clustering and principal component analysis revealed a more specific pattern for AdCy with low expression of the common gene signature. In situ studies at the RNA and protein level confirmed these results and indicated that, in contrast to MuEp and SaDu, ECM production in AdCy results from tumour cells and not from cancer-associated fibroblasts (CAFs). Our findings reveal different modes of ECM production leading to common and distinct RNA signatures in SGC. Of note, an overexpression of *COL27A1*, as in AdCy, has not been linked to any other neoplasm so far. Here, we contribute to the dissection of the ECM composition in SGC and identified a panel of deferentially expressed genes, which could be putative targets for SGC therapy and overcoming therapeutic resistance.

## 1. Introduction

The extracellular matrix (ECM) is an integral part of the tumour microenvironment (TME). It is composed of collagens, glycoproteins and matricellular proteins that modulate its composition or mediate the tumour cell–matrix interaction [1,2]. All matrix proteins have recently been termed “the matrisome” [3]. This organ- and context-specific network [4] is gradually altered throughout tumour initiation [5,6,7,8], progression and metastasis [9,10] as it provides a niche for tumour cell survival [4,11]. It is considered to be mainly secreted by cancer-associated fibroblasts (CAF) but tumour cells also seem to produce ECM molecules [12,13]. High ratios of the resulting peritumoural stroma have been associated with an adverse outcome in a multitude of primaries [14]. This effect can be partially attributed to ECM-stiffening through collagen cross-linking and ECM-deposition, which leads to augmented invasive capacities [15], epithelial-to-mesenchymal transition [16] and the reduced bioavailability of antineoplastic agents [17,18]. Moreover, individual ECM components can specifically enhance cancer cell dissemination and metastasis [2]. Although transcriptomic studies have recently demonstrated that certain ECM-signatures are prognostic of patient survival in several other primaries [19,20,21,22], the ECM composition of salivary gland carcinomas (SGC) remains largely unknown [23,24,25].

SGCs are a group of rare tumours with a wide range of growth patterns. Some carcinoma entities are multiphasic and, among others, consist of luminal, squamoid or myoepithelial cells [26]. This spectrum of different tumour cell features offers a valuable opportunity to study carcinoma-induced ECM changes as a function of cellular differentiation. Mucoepidermoid carcinomas (MuEp) are considered the most prevalent SGC subtype. They have a characteristic triphasic growth pattern (mucinous, squamoid and intermediate cells) and harbour gene fusions involving the *MAML2* gene [27]. Their architecture ranges from predominantly cystic (low grade) to solid (high grade) [28]. One of the histomorphological hallmarks of MuEp is a prominent fibrous stroma, which is considered a reactive change secondary to a leakage of intracystic mucus. Adenoid cystic carcinomas (AdCy) are defined as biphasic tumours with a luminal and at least a focal myoepithelial/basal differentiation. They have a well-defined genotype with a *MYB-NFIB* or *MYBL1-NFIB* translocation [29,30,31] and usually display an enigmatic cribriform or tubular growth pattern [26]. Conversely, salivary duct carcinomas (SaDu) are one of the few remaining salivary gland carcinoma subtypes without any known recurrent gene translocations [32,33]. As with other aggressive high-grade carcinomas, SaDu harbour sporadic mutations in well-recognised driver pathways [32,33] and are characterised by an overtly invasive growth and a prominent surrounding, desmoplastic stroma [34]. In contrast to the other two carcinoma types, they are monophasic. Although some SaDu exhibit a targetable androgen receptor expression [35] or *ERBB2* amplification [36], there is no specific therapy for most other SGC subtypes. Hence, new therapeutic approaches are required.

For the first time, we determined the expression of 28 ECM components in AdCy, MuEp and SaDu. These three entities cover the typical histomorphological and genetic features of SGC. Our results are a first step towards an entity-specific profiling of the matrisome in SGC. Knowledge of the ECM components of SGC will be crucial to reach the future goal of effectively targeting their TME.

## 2. Results

### 2.1. Patient Characteristics

The clinicopathological features of all 34 patients are displayed in Table 1. Specifically, 11 patients with AdCy, 14 with MuEp and 9 with SaDu were enrolled in the study. Overall, 63% of all cases were resected at a tumour stage pT3 or higher. Although most AdCy and SaDu had reached pT3 (72% and 88%), only 38% of all MuEp were resected at this or a higher stage. Nodal metastases were more frequent in SaDu (89%) than in AdCy (36%) or MuEp (31%). A female predilection was observed for AdCy and MuEp, whereas most SaDu cases occurred in males. At the time of diagnosis, patients with SaDu histology were older than patients with AdCy or MuEp (63 vs. 42 and 40 years, respectively).

### 2.2. Exploratory Data Analysis Revealed a Distinct ECM Profile of Adenoid Cystic Carcinomas

For this retrospective study, we extracted the expression counts for all available collagens, laminins and other ECM-related glycoproteins from the Nanostring PanCancer gene set (*n* = 28). Using normalised log_2_-fold values (tumour/normal tissue), the gene expression patterns were plotted as a heatmap with an adjacent dendrogram (Figure 1). Unsupervised hierarchical clustering revealed a cluster, which grouped 9 out of 11 AdCy together with two other non-AdCy cases. Cases with SaDu and MuEp histology as well as the remaining AdCy were not grouped according to their histological subtype. Furthermore, the heatmap revealed an overexpression of several genes, including *COL27A1* in the AdCy group, which formed a cluster.

This multidimensional dataset was further explored using principal component analysis (PCA). Principal components (PC) 1 and 2 accounted for 49.1% of the dataset’s variance. A first PCA plot was drawn to test if an age bias could have affected the expression patterns (Appendix A). All cases were split into two equally sized categories according to the date of sampling. This analysis demonstrated that both categories were largely overlapping, indicating that a different sample age did not lead to a shift in the overall gene expression. Next, we labelled all cases according to their histological diagnosis. The biplot in Appendix A illustrates both the samples’ positions and the different genes. Here, the different histological groups were separated by PC1 with a partially overlapping expression profile and MuEp was located between SaDu and AdCy. Again, expression of *COL27A1* separated AdCy from the other entities. Conversely, samples of SaDu were characterised by an expression of *COL1A1, COL1A2, COL5A1, COL3A1* and *FN1*.

### 2.3. The ECM of Salivary Duct and Mucoepidermoid Carcinomas Shared a Set of Mutually Overexpressed Genes

Our approaches revealed a number of genes that separated the histological groups by RNA expression. To investigate these relationships in detail, the log_2_-fold values of individual genes were compared. The 10 most overexpressed genes are displayed in Appendix A, whereas the *p*-values for all analysed genes can be observed in Appendix A. Boxplots for all collagens are displayed in Figure 2, whereas laminins and other ECM-related glycoproteins are shown in Appendix A, respectively.

Six genes (*COL1A1, COL27A1, COL5A1, COL5A2, COMP* and *LAMA5*) were significantly overexpressed in all three entities. No common downregulated genes were observed.

#### 2.3.1. Salivary Duct Carcinoma

In particular, 14 out of 28 genes (50%) were significantly overexpressed in the SaDu group compared to corresponding normal tissue (10 genes with log2-fold > 2). Precisely, 10 of these genes (36% of all genes) displayed a strikingly similar pattern of overexpression with an incremental increase from AdCy to MuEp and SaDu and at least a significant overexpression within the SaDu group. Among these, there were 6 collagens (*COL11A1*, *COL1A1*, *COL1A2*, *COL3A1*, *COL5A1* and *COL5A2*) and 4 glycoproteins (*COMP*, *FN1*, *SPP1* and *TNC*). In this context, significant differences between AdCy and SaDu were observed for 8 genes (*COL11A1*, *COL1A1*, *COL1A2*, *COL3A1*, *COL5A1*, *COL5A2*, *FN1* and *SPP1*). The incremental increase was most prominent in the case of fibronectin (*FN1*), with significant differences between all three entities.

Conversely, *LAMA1*, *LAMA3* and *LAMC3* were significantly downregulated in the group of SaDu compared to normal tissue (log_2_-fold > −2). Among these genes, the expression of laminin A1 was also significantly reduced compared to AdCy and MuEp.

#### 2.3.2. Mucoepidermoid Carcinoma

MuEp revealed an overexpression of 15 out of 28 genes (54%; 10 genes with log_2_-fold > 2). In particular, 9 of these genes (32%) shared the above-mentioned stepwise expression pattern, whereas only 6 genes were significantly upregulated compared to AdCy. In contrast, RNA expression of *COL24A1*, *LAMA1* and *LAMB3* was highest in MuEp.

The only significantly downregulated gene in MuEp was *LAMC3* (log2-fold > −2).

#### 2.3.3. Adenoid Cystic Carcinoma

In the AdCy group, only 11 of 28 genes (39%) were significantly overexpressed (5 genes with log_2_-fold > 2). Among these, only 4 genes (14%; *COL1A1, COL5A1 COL5A2* and *COMP*) followed the above-mentioned incremental gene expression pattern shared by SaDu and MuEp. Three collagens (*COL11A2, COL27A1* and *COL4A5*) were upregulated and showed higher values in AdCy than in the two other histological groups. In the case of *COL27A1*, the differences in expression between AdCy and the other two entities were statistically significant.

No genes were significantly downregulated in the AdCy group.

### 2.4. In Situ Validations Revealed ECM Production by Adenoid Cystic Tumour Cells

In order to validate the expression results of single relevant matrix-associated genes, we additionally performed in situ analyses (RNA and protein). This validation also allowed us to precisely determine in which cells, matrix genes (or proteins) are formed within the tumour. The in situ validations were performed for three exemplary cases of AdCy and two cases of each of MuEp and SaDu from the initial cohort. Additionally, we included 5 supplementary carcinomas of each tumour entity that were not analysed with the Nanostring method. As ECM production by tumour cells has also been reported [11,13], we used RNA in situ hybridisation (RNA-ISH) for *COL11A1* and *COL27A1* (Figure 3).

We chose *COL11A1* as a proxy for the mutual, incremental ECM expression pattern. It is a marker for cancer-associated fibroblasts (CAFs) and a component of general ECM signatures found in many epithelial neoplasms [37,38]. In keeping with our precedent results, *COL11A1* RNA-ISH stained numerous stromal cells in MuEp and SaDu. In the latter, we observed confluent RNA signals in cells compatible with CAFs (Figure 3). In contrast, we observed only few *COL11A1* positive CAFs in AdCy. The expression of *COL11A1* in AdCy could be almost exclusively attributed to the tumour cells (Figure 3). Fitting with the RNA-ISH results, the collagen 11A1 protein revealed a strong stromal staining in SaDu and MuEp and a very weak staining in the AdCy tumour matrix in a few cases (Appendix A).

As *COL27A1* expression was significantly higher in AdCy than in the other two entities, we further investigated its RNA expression in situ. For AdCy, we noted a consistent, intense staining in >95% of tumour cells, with confluent signals in the RNA-ISH. In normal peritumoural salivary gland tissue, single signals were frequently noted in the ductal epithelium and lymphatic tissue. MuEp and SaDu both had focal areas with single signals in most tumour cells, whereas a few SaDu cases revealed a more prominent staining.

LAMB3 protein was diffusely expressed in the tumour cells of most MuEp cases. Staining of peripheral cells was noted in SaDu specimens. A few SaDu cases also revealed a diffuse staining in areas where a growth pattern with small solid tumour clusters was observed. Peripheral cells of tumour cell groups were also stained in some AdCy cases. In peritumoural salivary tissues, no staining of LAMB3 protein was detected.

### 2.5. MALDI-TOF-MS-Imaging (MSI) Confirmed Specific Upregulation of COL27A1 in AdCy at the Protein Level

Our RNA-expression results demonstrated a specific upregulation of *COL27A1* in AdCy. To confirm this differential expression at the protein level in an unbiased, quantitative manner, we performed MSI on two representative tissue microarray (TMA) slides that contained AdCy and various other SGC subtypes. As indicated in Figure 4, we were able to demonstrate the expression of COL27A1 protein, predominantly in AdCy. The in silico digestion of COL27A1 produced 531 peptides. Out of this list, 262 m/z values were identified in the TMA sample cohort (Appendix A). These m/z values could be correlated with 88 peptides from COL27A1 due to the fact that with MALDI several adducts from the same peptide can arise (e.g., H^+^, Na^+^ and NH_4_^+^). We detected a significantly higher expression of COL27A1 in AdCy than in the other carcinoma entities (*p*-value < 0.001). This allowed a confident discrimination between AdCy and the other entities by COL27A1 protein detection and largely confirmed our results from the RNA expression analysis.

## 3. Discussion

Here, for the first time, we describe the differential matrisome of three prototypic carcinoma entities of the salivary gland. We used a panel of 28 genes, including collagens, laminins and glycoproteins, to describe mutual and exclusive features of their respective ECM. A common, desmoplastic stromal reaction, with overexpression of 10 genes (36%) was detected in SaDu and MuEp. Conversely, AdCy displayed a more distinct ECM pattern.

Incremental increase in expression from AdCy over MuEp and SaDu samples was observed for 6 out of 11 collagens and for 4 out of 6 nonlaminin glycoproteins. This upregulation in MuEp and SaDu is unlikely to correspond to specific neoplastic features as both entities bear contrasting immunohistochemical phenotypes and have profound pathogenetic differences. Hence, these genes might be part of a mutual, desmoplastic stromal reaction, which differs only quantitatively in MuEp and SaDu. Such converging changes in the ECM are observed in multiple carcinoma types. Lim et al. developed an ECM-related gene set, which had prognostic value for some of the most prevalent carcinoma types, including lung, colon and breast carcinomas. Among these, there were three genes, which also appeared in our mutually overexpressed set of 10 genes (*COL11A1*, *SPP1* and *TNC*), indicating that SGC might partially share common carcinoma-induced ECM changes [19,20]

We observed abundant *COL11A1*+ stromal cells in SaDu samples and to a lesser extent in MuEp. Jia et al. demonstrated that *COL11A1*+ stromal cells in carcinomas are *COL11A1*+ CAFs [38]. The expression of collagen 11A1 (*COL11A1)* by CAFs in several carcinoma types, including breast [39], kidney [40], ovarian [41,42,43] and colon adenocarcinoma [44,45], has been associated with an adverse outcome. This might be explained by the evidence that *COL11A1* promotes the invasion and migration ability of carcinoma cells and is involved in the formation of metastases [46].

As part of a positive feedback loop, deposition of ECM components leads to a stiffened matrix, resulting in decreased perfusion and tissue hypoxia [47,48]. We demonstrated that several genes which are involved in widespread desmoplastic signatures [20,38] are also overexpressed in SaDu and MuEp. Furthermore, we observed a high density of *COL11A1*+ CAFs along with a strong *COL11A1* upregulation in these two histological groups. Thus, agents that target CAFs or modulate stromal stiffness and hypoperfusion might also be applicable for these two carcinoma types [18,49].

MuEp, which displayed a partial squamoid differentiation, revealed the highest expression of the laminin subunit B3 (*LAMB3*). Furthermore, we also observed an overexpression of laminin subunit C2 (*LAMC2*) in MuEp, as well as in AdCy. Physiologically, *LAMB3* is part of the heterotrimer laminin 332 (laminins A3, B3 and C2) and plays a pivotal role in the context of cell adhesion. Among several other carcinomas, it is expressed in head and neck squamous carcinomas [50] and promotes tumour growth [51,52,53]. In addition, the overexpression of subunits of Laminin 332 has been associated with an adverse outcome in several primaries [54,55,56,57]. Interestingly, we could not detect an overexpression of *LAMC2* in MuEp. To our knowledge, such an isolated overexpression of *LAMC2* and *LAMB3* has not been described in any carcinoma type.

Exploratory data analysis revealed that AdCy cases form a distinct cluster. The gene expression pattern which was shared by SaDu and MuEp was expressed to a much lesser extent in AdCy, and collagen 27A1 (*COL27A1)* was upregulated in AdCy, with significantly lower expression levels in the other salivary gland carcinomas. This notion was further supported by the complementary MSI analyses that we performed to detect collagen 27A1 protein on two representative TMA slides. *COL27A1* plays a role in the development of murine and human cartilage and bone [58,59]. More recently, mutations in the *COL27A1* gene have been associated with Steel syndrome, a genetic disorder leading to bone malformation [60,61]. To our knowledge, we describe for the first time an upregulation of *COL27A1* in a human cancer. Future studies should address its functional role in AdCy. The abundant expression of *COL27A1* RNA and protein in AdCy as well as the highly restricted physiological expression of this collagen might provide a rationale to explore *COL27A1* as a potential target.

Matrix protein production by tumour cells has been noted in other primaries [12,13,62]. Recently, ECM protein expression in pancreatic ductal carcinoma cells, rather than in stromal cells, has been shown to correlate with poor prognosis [13]. Surprisingly, in case of AdCy, we hardly noticed any *COL11A1*+ CAFs in the TME, which is in contrast to the abundance of CAFs in the other two entities. Conversely, RNA expression of the two exemplary collagens (*COl27A1* and *COL11A1*) was nearly restricted to tumour cells in AdCy. This might explain the very low expression of the mutual ECM pattern found in MuEp and SaDu. Furthermore, our results are in keeping with other reports of ECM production by adenoid cystic carcinoma cells [63] and myoepithelial cells in general [64]. AdCy tumour cells might be the main source for ECM proteins instead of CAFs, which marks a profound difference to the other tumour entities. Considering the complex interplay of CAFs and tumour cells, this finding emphasises that future studies should determine whether the production of targetable ECM components is attributed to stromal or tumour cells.

Several ECM components such as fibronectin (*FN1*) [65], tenascin C (*TNC*) [66], thrombospondin 1 (*THBS1)* [67] and osteopontin (*SPP1*) [68], which were highly overexpressed in our dataset, are associated with an adverse prognosis and represent potential therapeutic targets [2,65,66,67,68]. Unfortunately, many ECM-related proteins are not exclusively expressed in cancer tissues, making direct targeting difficult. On the other hand, their abundant expression in carcinomas makes them a promising tool to direct established agents to the tumour, which may decrease otherwise severe side effects [69]. *FN1* was among the 10 most overexpressed genes in the group of SaDu and MuEp. Several studies have assessed the therapeutic potential of the fusion protein L19-IL2, which consists of an interleukin 2 (IL2) coupled to a monoclonal antibody directed against *FN1*. Promising experimental and clinical results have been reported in case of metastatic melanoma [65] and renal cell carcinoma [70]. SaDu tumours with a high tumour mutational burden and a high amount of immune infiltration might be eligible for therapy with L19-IL2 due to their high immunogenicity [34,71].

The deposition of several ECM components, such as collagens, laminin 332 and *FN1*, can lead to a resistance against chemotherapy [72]. A well-established concept is that a deposition and cross-linking of ECM components such as collagen 1 lead to increased matrix stiffness, which in turn reduces perfusion with antineoplastic agents. These mechanisms can be therapeutically addressed by molecules that target the renin–angiotensin system [18,73,74]. *COL1A1* and *COL1A2* were significantly overexpressed in MuEp and SaDu. Future studies should address a possible targeting of collagen 1 in SGC to overcome stiffness-induced chemoresistance. Another example of collagen-induced chemoresistance is the upregulation of fatty acid oxidation (FAO) through CAF-derived *COL11A1*, leading to cisplatin resistance [75]. As we observed a prominent upregulation of *COL11A1* in MuEp and SaDu, *COL11A1*-induced FAO could be another future therapeutic target for SGC.

Our retrospective study has some limitations which are due to its exploratory character. First, we analysed only 28 ECM-related genes, which are included in the Nanostring PanCancer gene set. Further studies should address all genes that are part of the matrisome. Second, instead of including all different histological entities in this study, we focused on three enigmatic carcinoma types that cover the histological and genetic hallmarks of salivary gland tumours. Third, the relatively small number of cases per entity does not allow for prognostic analyses or subgrouping with regard to clinicopathological parameters. Subsequent study designs should include a larger number of cases and correlate the different ECM components with clinical outcomes. Fourth, although we included normal salivary gland tissue from each patient as a reference, we did not analyse tissue from normal controls. Thus, we cannot exclude that underlying mechanisms of the disease might have had an impact on the control tissue (e.g., germline mutations).

In summary, we revealed an overexpression of ECM components, which are associated with enhanced migratory capacities of tumour cells (*TNC* [76,77], *FN1* [78]), resistance to chemotherapy (*COL11A1* [43,75], *SPP1* [79]) and adverse outcome (*COL11A1* [42], *TNC* [80], *FN1* [78]) in other primaries. The functional role of these molecules and their potential as therapeutic targets are currently under intense investigation [65,66,75,79,81,82]. Hence, as new therapeutic approaches for SGC are needed, our findings are of high translational relevance. Future studies should address the functional role of *COL1A1*, *COL11A1*, *FN1*, *TNC*, *THBS1* and *SPP1* in SGC to evaluate a potential clinical targeting.

We used a wide spectrum of salivary gland carcinomas to gain insights into the differential matrisome of carcinomas with varying cellular differentiation. Our findings provide a strong rationale to study the matrisome of carcinomas based on their histomorphological classification. Furthermore, our results encourage a detailed profiling of ECM components at a cellular level in situ to gain further insights into the tumour matrisome.

## 4. Materials and Methods

### 4.1. Patient Characteristics

We searched the archives of the Institute of Pathology, University of Cologne for AdCy, MuEP and SaDu resection specimens. Cases for which there was sufficient FFPE material and informed consent had been given were passed on to histopathological revision. Here, inclusion criteria were defined as follows: salivary gland carcinoma of the parotid gland; unequivocal histopathological diagnosis of either AdCy, MuEp or SaDu and sufficient FFPE material of tumour and normal tissue as defined by the manufacturer’s requirements. All histological diagnoses were reviewed by two pathologists with special expertise in the field (C.A. and A.Q.) using a panel of the following antibodies: CK7, p63, S100, androgen receptor, HER2 and NOR-1. In addition, in cases of remaining diagnostic uncertainty, only cases with either *MYB/MYBL1* or *MAML2* translocation, as indicated by break apart FISH, were designated as AdCy or MuEp, respectively. After this workup, all cases with an ambiguous diagnosis were strictly excluded from the study. TNM classifications were corrected according to the latest WHO classification of head and neck tumours (8th edition, 2020). All 34 SGC patients included in the study underwent primary surgery at the Department of Otorhinolaryngology, Head and Neck Surgery, University Hospital of Cologne, Germany between 1990 and 2014. The investigation was conducted according to the Declaration of Helsinki on biomedical research involving human subjects. The study was approved by the Ethics Committee of the University of Cologne (13-265).

### 4.2. Immunohistochemistry

Briefly, tissue slides were stained with antibodies against COL11A1 (rabbit polyclonal, 1:200, Biozol, Eching, Germany) and LAMB3 (mouse monoclonal, clone OTI2D11, 1:1000, OriGene, Rockville, MD, USA). All IHC stainings were performed using a Leica BOND-MAX stainer (Leica Biosystems, Wetzlar, Germany) in accordance with the manufacturer’s protocol. Counterstaining was carried out with haematoxylin and bluing reagent.

### 4.3. RNA In Situ Hybridisation (RNA-ISH)

The RNAscope assay was carried out in accordance with manufacturer’s instruction, as previously described [83]. In brief, 5 μm sections were cut from FFPE blocks, pretreated according to an extended protocol (30 min for pretreatment 2 and 3), digested and hybridised at 40 °C with human *COL11A1* or *COL27A1* mRNA probes (both ready to use, Advanced Cell Diagnostics, Bio-Techne, Minneapolis, MN, USA) using a Ventana Discovery system (Roche, Switzerland). Afterwards, the slides were incubated for 10 s in haematoxylin.

### 4.4. RNA Processing

Tumour areas with at least 70% tumour cell count as well as corresponding normal tissue were marked on H&E slides. After scraping off the respective tissue of the FFPE blocks, RNA isolation was carried out using the RNA FFPE Kit from Promega on the Maxwell^®^ 16 LEV (Promega AS1260, Heidelberg, Germany), according to the manufacturer’s instructions. RNA concentrations were quantified by the QuantiFluor^®^ RNA System as recommended by the supplier (Promega, E6090) and adjusted to 30 ng/µL. The nCounter Nanostring analysis was carried out with 150 ng RNA according to the manufacturer’s instructions. The PanCancer Pathways Panel (Nanostring, Seattle, WA, USA) was used following the manufacturer’s recommendations. In brief, the RNA was denatured at 85 °C, followed by incubation with both probes (reporter and capture probe) for 12 h at 65 °C. Afterwards, the barcoded hybrids were purified with the Prepstation platform and quantified using the Digital Analyzer (both Nanostring, Seattle, WA, USA).

### 4.5. Data Processing

Raw expression counts of tumour samples and normal tissue were imported into the nSolver 4.0 Software (Nanostring, Seattle, WA, USA) for quality control and normalisation procedures. All but one sample passed the in-build quality controls and were passed on to further processing (*n* = 34). Normalisation was done using the geometric mean of the CodeSet Content housekeeping genes and positive controls. The background noise threshold was set to the maximum of all negative controls. All values below a background noise threshold were floored to this value. After an extensive literature research of ECM molecules, we selected all collagens, laminins and other ECM-related glycoproteins from the PanCancer gene set for further analysis (*COL11A1*, *COL11A2*, *COL1A1*, *COL1A2*, *COL24A1*, *COL27A1*, *COL2A1*, *COL3A1*, *COL4A3*, *COL4A4*, *COL4A5*, *COL4A6*, *COL5A1*, *COL5A2*, *COL6A6*, *LAMA1*, *LAMA3*, *LAMA5*, *LAMB3*, *LAMB4*, *LAMC2*, *LAMC3*, *COMP*, *FN1*, *SPP1*, *TNC*, *THBS1* and *THBS4*).

### 4.6. Statistical Analysis

The statistical analyses were performed with R Studio (version 1.2.5033, RStudio, Boston, MA, USA). First, normalised counts of all cancer samples and normal tissue probes were each pooled entity wise and tested for differential expression using Wilcoxon tests (rstatix package), and the *p*-values were adjusted for multiple testing with the Benjamini-Hochberg procedure (FDR < 0.05). Second, the normalised counts for cancer and normal tissue of each patient were used to build log_2_-fold ratios for each gene and patient. These ratios were then tested for interentity differences and visualised as boxplots (ggplot2 package). Symbols of significance levels were manually pasted in Figure 2, Appendix A using Microsoft PowerPoint for Mac (version 16.35, Microsoft Corporation, Seattle, WA, USA). For the exploratory analyses (cluster analysis, PCA), all log_2_ were scaled and centred. The cluster analysis was carried out and illustrated with the hclust (stats package) and pheatmap function (pheatmap package). Clustering was performed with the “ward.d2” method, which uses Ward’s hierarchical clustering method (agglomerative hierarchical clustering). The functions prcomp (stats package) and ggbiplot (ggbiplot package) were used to carry out and visualise the PCA as biplot.

### 4.7. MALDI-TOF-MS-Imaging (MSI)

TMA sections were mounted on indium tin oxide (ITO)-coated glass slides. Sample preparation was performed according to previously published protocols [84]. In brief, the following steps were done: deparaffinisation, washing with xylene and ethanol, pH conditioning and antigen retrieval. For on-tissue enzyme digestion, a trypsin solution of 2 µg/µL trypsin (Promega Gold) in 50 mM AMBIC buffer and 10% ACN was used. As spraying device, the SunCollect System (SunChrom, Friedrichsdorf, Germany) was used. Digestion was performed with the SunDigest chamber (SunChrom, Friedrichsdorf, Germany) in basic mode. The matrix used for the experiments consisted of 10 mg/mL CHCA in 60% ACN and 0.2% TFA and was applied with the SunCollect.

MSI data acquisition was performed using the UltrafleXtreme mass spectrometer (Bruker Daltonik, Bremen, Germany) with the following parameters: spatial resolution of 50 µm, medium laser spot size, 200 shots, mass range from 500 to 4000 m/z and in positive ion reflector mode. The Bruker peptide standard was spotted separately on the ITO slide as external calibration. After MSI measurements, the sections were washed and stained with haematoxylin and eosin. The slides were scanned using the NanoZoomer-SQ slide scanner (Hamamatsu Photonics, Hamamatsu, Japan).

For data and imaging analysis, the SCiLS Lab software (Bruker Daltonik, Bremen, Germany), version 2020b Premium 3D, was used. All data were TIC normalised. A baseline removal algorithm convolution with peak width 20 was used. For detection of m/z values, which could be correlated to COL27A1, in silico digestion was performed using the ProteinProspector MS-Digest tool. For the database search, UniProtKB.2017.11.01 was used with Q8IZC6 as the UniProt-ID. The segmentation pipeline in SCiLS was performed with the in silico digestion m/z list. For statistical analysis, the Wilcoxon test was carried out.

### 4.8. Data Availability

*p*-values and log_2_-fold values for differential expression analyses are listed in Appendix A. Normalised counts for all genes that were addressed in the manuscript can be accessed at https://www.ncbi.nlm.nih.gov/geo/info/linking.html (record: GSE153283). All detected peptides from MSI analyses are listed in Appendix A.

## 5. Conclusions

The ECM of carcinomas is vital for cancer progression and therapy resistance. The heterogenic group of SGCs exhibits a diverse cellular and stromal differentiation, thus representing an ideal model to study differential ECM expression and secretion. For the first time, we depicted the ECM composition of the three most enigmatic salivary gland carcinomas in a quantitative and multidimensional manner at the RNA level. We revealed that MuEp and SaDu share a common CAF-derived stromal signature, in contrast to AdCy, which displays a distinct, tumour cell-derived ECM, possibly due to its partial myoepithelial differentiation. Our results lead to the following main conclusions: (1) The ECM of SGC is highly diverse and provides several potential therapeutic targets and (2) Although carcinomas with monophasic luminal differentiation (MuEp, SaDu) seem to share common stromal signatures with other primaries, myoepithelial differentiation, as in AdCy, might confer a distinct, tumour cell-based ECM.

Our findings contribute to the elucidation of ECM alterations in SGC, a process that will be crucial to achieve the future goal of overcoming therapy resistance. Together, we provide a rationale to address the ECM as a function of cellular differentiation.

## Figures and Tables

**Figure 1 cancers-12-02466-f001:**
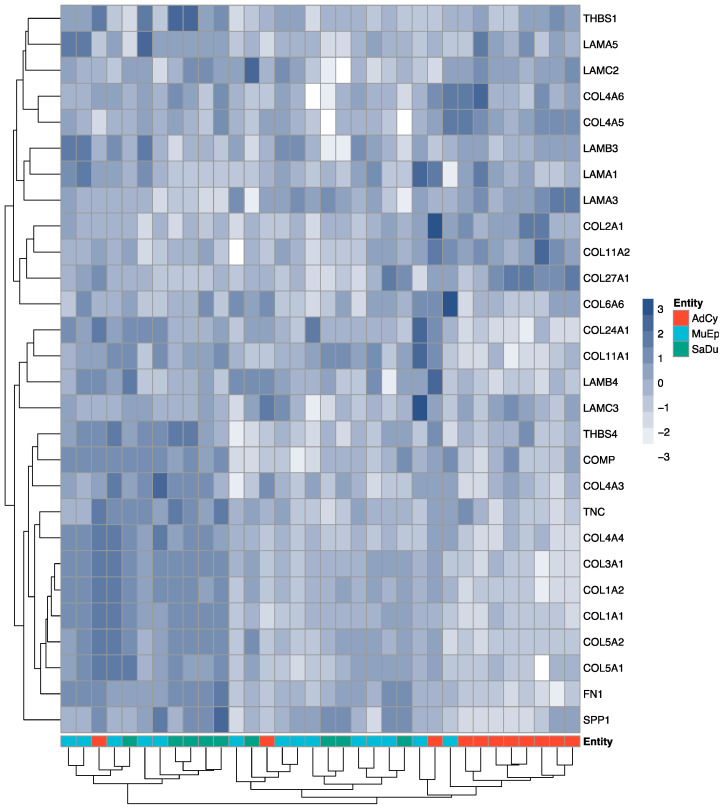
Gene expression heatmap. RNA expression log_2_-fold values for each patient (x-axis) and each gene (y-axis) are displayed from +3 (dark blue) to −3 (white). At the bottom row, the respective histologic groups are shown. Hierarchical clustering of samples and genes (dendrogram) was performed using Ward’s method. AdCy: adenoid cystic carcinoma; MuEp: mucoepidermoid carcinoma; SaDu: salivary duct carcinoma.

**Figure 2 cancers-12-02466-f002:**
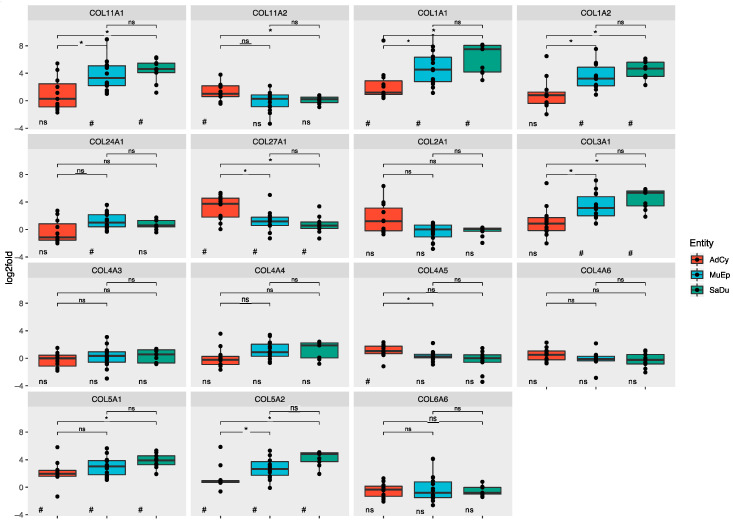
Boxplot depicting entity-wise RNA expression of all analysed collagens s log_2_ fold values. Individual cases are displayed as points. Significant differences for comparisons between two individual histologic groups are given as asterisks above the boxplots (adjusted *p*-values; *: *p* < 0.05; ns: not significant). Significance levels for comparisons between tumour and normal tissue are given as number symbols below the boxplots (adjusted *p*-values; #: *p* < 0.05; ##: *p* < 0.01; ###: *p* < 0.001; ns: not significant). AdCy: adenoid cystic carcinoma; MuEp: mucoepidermoid carcinoma; SaDu: salivary duct carcinoma.

**Figure 3 cancers-12-02466-f003:**
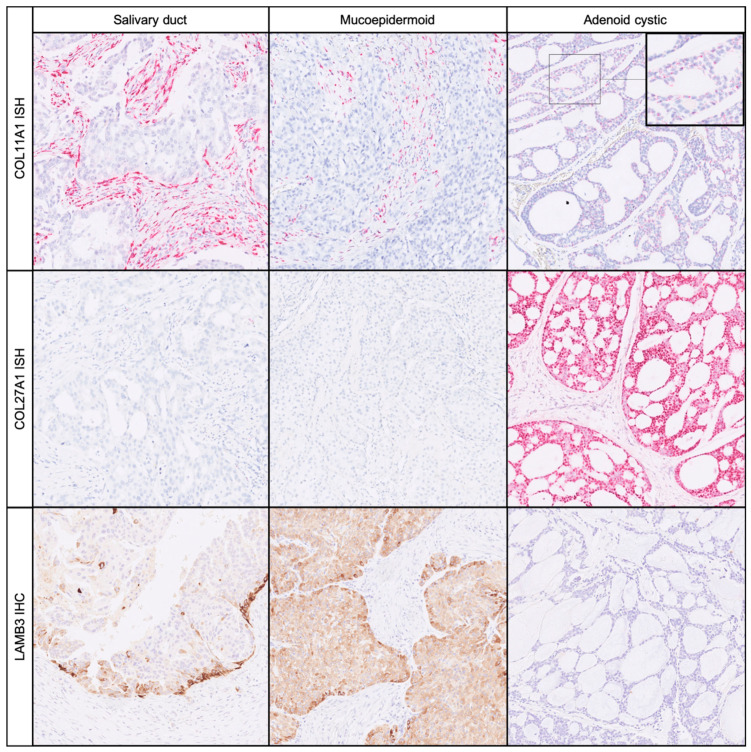
In situ validation of selected extracellular matrix (ECM) gene expressions by in situ hybridisation (ISH) and immunohistochemistry (IHC). Crosstab demonstrating the expression of *COL11A1* RNA (top row), *COL27A1* RNA (middle row) and laminin subunit B3 (*LAMB3*) protein (bottom row) for salivary duct, mucoepidermoid and adenoid cystic carcinoma (columns, from left to right); magnification: 100× (inset 200×).

**Figure 4 cancers-12-02466-f004:**
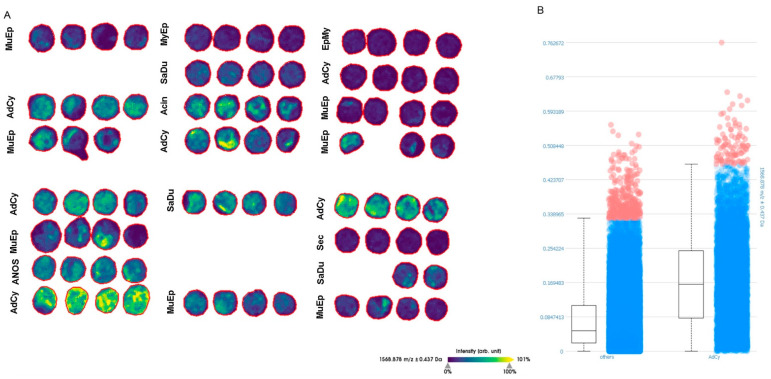
Distribution of one COL27A1 peptide in representative tissue microarray (TMA) sections. (**A**) Colour-coded intensity map of 1586.878 m/z measured with MALDI MS-imaging (MSI), corresponding to m/z 1545.8166 [M+ Na+] with the peptide sequence (K)MDRGDGLKTGVQLR(K) and two trypsin miss cleavages from in silico digestion. In five out of six patients with AdCy, an expression of COL27A1 peptide was detected. (**B**) Intensity box plot comparing the intensity of the m/z value of AdCy and all other carcinoma entities. Wilcoxon rank-sum corrected *p*-value < 0.001. AdCy: adenoid cystic carcinoma; MuEp: mucoepidermoid carcinoma; SaDu: salivary duct carcinoma; MyEp: myoepithelial carcinoma; EpMy: epithelial-myoepithelial carcinoma; Acin: acinic cell carcinoma; Sec: secretory carcinoma, ANOS: adenocarcinoma not-otherwise-specified.

**Table 1 cancers-12-02466-t001:** Overview of all included samples organised by histologic group. Percentages per entity are given in brackets.

Clinicopathological Parameter		AdCy (*n* = 11)	MuEp (*n* = 14)	SaDu (*n* = 9)	Total (*n* = 34)
Grade					
	1	1 (9)	8 (57)	5 (56)	14 (41)
	2	5 (46)	2 (14)	4 (44)	11 (32)
	3	3 (27)	4 (29)	0 (0)	7 (21)
	NA	2 (18)	0 (0)	0 (0)	2 (6)
pT					
	1	2 (18)	2 (15)	0 (0)	4 (12)
	2	1 (9)	6 (46)	1 (11)	8 (24)
	3	2 (18)	2 (15)	4 (44)	8 (24)
	4a	3 (27)	1 (8)	3 (33)	7 (21)
	4b	1 (9)	2 (15)	1 (11)	4 (12)
	NA	2 (18)	0 (0)	0 (0)	2 (6)
pN					
	0	6 (55)	9 (69)	1 (11)	16 (49)
	1	3 (27)	0 (0)	1 (11)	4 (12)
	2	1 (9)	4 (31)	7 (78)	12 (36)
	NA	1 (9)	0 (0)	0 (0)	1 (3)
L					
	L0	7 (70)	9 (82)	6 (67)	22 (73)
	L1	0 (0)	2 (18)	3 (33)	5 (17)
	NA	3 (30)	0 (0)	0 (0)	3 (10)
V					
	V0	7 (70)	8 (73)	8 (89)	23 (77)
	V1	0 (0)	3 (27)	1 (11)	4 (13)
	NA	3 (30)	0 (0)	0 (0)	3 (10)
Pn					
	Pn0	5 (46)	6 (50)	4 (44)	15 (47)
	Pn1	3 (27)	6 (50)	5 (56)	14 (44)
	NA	3 (27)	0 (0)	0 (0)	3 (9)
Sex					
	f	8 (73)	10 (71)	4 (44)	22 (65)
	m	3 (27)	4 (29)	5 (56)	12 (35)
Age, years					
Median (range)		42 (30–73)	40 (16–68)	63 (37–78)	46 (16–78)

AdCy: adenoid cystic carcinoma; MuEp: mucoepidermoid carcinoma; SaDu: salivary duct carcinoma; NA: information not available; L: lymphatic invasion; V: vascular invasion; Pn: perineural invasion; f: female; m: male.

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
