# Peer review of "Expression Profiling of Extracellular Matrix Genes Reveals Global and Entity-Specific Characteristics in Adenoid Cystic, Mucoepidermoid and Salivary Duct Carcinomas"

_cancers, 2020, doi:10.3390/cancers12092466_

Round 1

Reviewer 1 Report

This is a very unique study on ECM of salivary gland cancers. The authors examined RNA expression levels of ECM related genes based on Nanostring assay. This represents a novel finding and certainly deserves to be presented, but I do have a few comments/questions:

  1. Please clarify the list of genes examined for RNA expression in method section.
  2. The authors provided IHC of COL11A1 and LAB3. Is there an antibody for COL27A1? It might be worthwhile to prove that expression pattern is different between AdCY versus the others.
  3. How old were the specimens? It is known that quality of RNA depends on the age of FFPE specimens. Is there any possibility of systematic bias based on differences in age of specimens among different tumor type?
  4. In conclusion, the authors claim that difference in ECM may confer treatment resistance, but there is no supporting data on it. I would tone down, unless there is a convincing supporting evidence for the statement. 

Author Response

Comment 1: Please clarify the list of genes examined for RNA expression in method section.

Response: A very helpful remark. We have listed all 28 analyzed genes in the method section sorted by gene group (collagens, laminins, other glycoproteins, lines 423-426).

Comment 2: The authors provided IHC of COL11A1 and LAB3. Is there an antibody for COL27A1? It might be worthwhile to prove that expression pattern is different between AdCY versus the others.

Response: We tried to establish and calibrate an anti-COL27A1 antibody. Unfortunately, none of the two commercially available clones we ordered produced a satisfactory staining, neither on positive controls (e.g. hyaline cartilage, other chondroid tissue) nor on AdCy. Nonetheless, we insisted on a validation for COL27A1 protein. Fortunately, the group of Prof. v. Eggeling provided us with the mass spectrometry imaging technology. On the one hand, this method does not reach a subcellular resolution as achieved with IHC. On the other hand, it allowed for a quantitative analysis of the protein and revealed that also COL27A1 protein was significantly overexpressed in AdCy compared to the other tumor types. Overall, we found that this approach was well-suited to validate the Nanostring results even though it lacks a high spatial resolution. We hope that in the future, reliable anti-COL27A1 antibodies will be provided by the suppliers.

Comment 3: How old were the specimens? It is known that quality of RNA depends on the age of FFPE specimens. Is there any possibility of systematic bias based on differences in age of specimens among different tumor type?

Response: Thank you for this valuable remark. As indicated in the manuscript (line 376-379), the specimen have been resected between 1990 and 2014. As suggested, to address the possibility of an age bias, we have split all cases into two equally sized groups according to the date of surgery. Now, a first PCA plot (Supp. Fig. 1) demonstrates that samples from both categories diffusely overlap, indicating that different sample age did not lead to a shift of the overall gene expression (lines 112-118). In general, the Nanostring system is a very robust method to quantify RNA from FFPE tissue (Technote: https://www.nanostring.com/scientific-content/technology-overview/ncounter-technology/ffpe-simplified). Apart from all target genes, also endogenous housekeeping genes, as well as in-build positive and negative controls are measured separately for each sample. Using these genes, we normalized the expression counts for all samples according to the manufacturer’s instructions (nSolver 4.0 software). This compensates a possible loss of RNA through age-related degradation and practically excludes a technical inter-sample bias. Moreover, all samples met the required amount of input RNA as specified by the manufacturer (lines 404-414).

Comment 4: In conclusion, the authors claim that difference in ECM may confer treatment resistance, but there is no supporting data on it. I would tone down, unless there is a convincing supporting evidence for the statement. 

Response: We address this issue now to a larger extend in the discussion (lines 321-331). Indeed, a plethora of studies have depicted how ECM components and CAFs can mediate to chemoresistance (reviewed in: doi:10.3390/ijms18071586; doi:10.3390/cancers7040902; https://doi.org/10.1186/s12967-019-2058-1). Underlying mechanisms of chemoresistance for several individual proteins have been comprehensively analyzed in functional studies. As COL11A1 and COL1A1/COL1A2 were significantly overexpressed in our dataset, we now discuss such mechanisms that involve these two collagens (lines 325-331). We believe that we now provide sufficient data to support the conclusion that certain ECM molecules can promote chemoresistance.

Reviewer 2 Report

Arolt et al. present a quantitative analysis on mRNA level of 28 ECM-related genes of 34 adenoid cystic (AdCy), mucoepidermoid (MuEp) and salivary duct carcinomas (SaDu). Confirmatory in-situ studies on RNA and protein level are provided.

Six collagens (including COL11A1) and four glycoproteins are found to be overexpressed upon comparison from MuEp to SaDu.

AdCy and MuEp displayed a distinct overexpression of COL27A1 and LAMB3, respectively.

Of further interest, the authors find, that in contrast to MuEp and SaDu, ECM production in AdCy results from tumour cells and not from CAFs.

The topic is interesting and the manuscript reports on new and partially insightful data. At times the wording seems a bit inconvenient. The manuscript could benefit substantially from language editing by a native speaker as one can only guess at the authors meaning on numerous occasions.

"...While transcriptomic studies have recently demonstrated certain ECM signatures that are prognostic of patient survival [16–18] for of several carcinoma types, the ECM composition of salivary gland carcinomas (SGC) remains unknown..."

Unfortunately, there are several major issues which prohibit acceptance of the manuscript in its current form and necessitate major revisions before publication can be further considered.

Major comments:

- The introduction needs to be rephrased as it is rather lengthy yet lacks references for several statements and an overall common thread. Currently, it reads more like several loosely interconnected paragraphs.

- The study group is rather small, severely limiting the validity of the results obtained by the authors.

- The PanCancer Pathways Panel (Nanostring) provides but a limited view of the ECM. Therefore, not the matrisome but rather a minor aspect of the latter is investigated. This needs to be clarified throughout the text.

- The observations should be supplemented by an investigation of the cellular composition of the TME (T-cell subsets, tumour-infiltrating macrophages).

- The discussion is too long and needs to be edited. Recurrently, the authors discuss observations and interpretations presented by other investigators that are only loosely connected to the current data-set.

- The retrospective nature of the study is another critical point as well as the extensive period of recruitment. Gene expression data need to be investigated for a sample-age related bias.

- Clinical data lack follow-up. The study is merely descriptive in nature and provides no clinical correlate. Expanding the study into a multi-centre setting might enable meaningful interpretation of TME findings in a clinical context.

- Validation in an independent data-set should be performed. If possible, as part of the suggested multi-institutional approach.

- Quantitative PCR validation should be considered for all significant genes.

- Histopathological work-up of the study group in insufficiently depicted. Especially molecular studies are not reported (e.g. MYB-NFIB/MYBL1-NFIB/MAML2 associated translocations or fusions)

Minor comments:

- Please provide an explanation for every abbreviation upon first use (CAF).

- Boxplots are unsuitable for the depiction of datasets as small as this one (e.g. n = 9). Individual datapoint should be plotted instead.

Author Response

Comment on English language: The manuscript could benefit substantially from language editing by a native speaker as one can only guess at the authors meaning on numerous occasions.

Response: We critically reviewed wording and grammar. Additionally, we had a professional editing service edit the entire manuscript. We hope that this improved the linguistic quality to a large extend.

Major comments:

Major comment 1: The introduction needs to be rephrased as it is rather lengthy yet lacks references for several statements and an overall common thread. Currently, it reads more like several loosely interconnected paragraphs.

Response: Thank you for this valuable contribution. As suggested, we shortened the introduction significantly. Also, the introduction now consists of three interconnected parts. Part one is an introduction to the ECM and its clinical implications. In the following part, a brief introduction to the studied SGC subtypes is given. Third, we give a short outlook on the study design. We hope that these changes improve overall readability and provide a sufficient, yet short introduction to the overarching topic. Moreover, we now provide comprehensive references for all aspects covered in the introduction. 

Major comment 2: The study group is rather small, severely limiting the validity of the results obtained by the authors.

Response: We accept this justified criticism. As this work is designed to be a pilot study, covering three enigmatic SGC entities and some of the best characterized ECM components, we aimed to provide a substantial impression on the ECM of SGC. However, as you remarked, a larger cohort would substantially increase the study’s power. Unfortunately, it remains impossible for us to extend the study cohort or even recruit another cohort from a different center within the timeframe specified by the editor. Such a study design would require at least another year of time and substantially more financial resources to establish. The entirety of the matrisome and a more extensive cohort will be addressed by us in future studies.

Major comment 3: The PanCancer Pathways Panel (Nanostring) provides but a limited view of the ECM. Therefore, not the matrisome but rather a minor aspect of the latter is investigated. This needs to be clarified throughout the text.

Response: Thank you for this comment. Now, we state in all parts of the manuscript that we analyzed only 28 ECM-related genes (lines 2, 26, 80, 104, 148, 245, 283, 347). Moreover, we address the selection of these genes in the results (lines 103-104) as well as in the M&M section (lines 421-426) and state this limitation of the study within the discussion (346-347). Also, we changed the title of the manuscript, indicating that we neither analyzed the entire matrisome nor all SGC subtypes (lines 2-4).

Major comment 4: The observations should be supplemented by an investigation of the cellular composition of the TME (T-cell subsets, tumour-infiltrating macrophages).

Response: As proposed, we investigated the frequency of tumor infiltrating CD8+ cytotoxic T cells, CD4+ T helper cells and CD68+ macrophages. Using statistical tests, we compared the respective frequencies between the tumor types.  Accordingly, the M&M section (lines 387-395, 441, 442), results (lines 232-242) and discussion (lines 333-344) have been updated.

Major comment 5: The discussion is too long and needs to be edited. Recurrently, the authors discuss observations and interpretations presented by other investigators that are only loosely connected to the current data-set.

Response: Thank you for this valuable criticism. As you suggested, we significantly shortened the discussion. We excluded details that were not directly linked to our results and hampered overall readability. Through these changes, the discussion has gained a stronger focus on our overall conclusions. Another reviewer suggested that we should discuss mechanisms of ECM-derived chemoresistance in more detail. In order to do this, we implemented a paragraph about this topic. Here, we only address ECM molecules that were significantly overexpressed in our dataset and could hence be relevant in SGC (lines 325-331).

Major comment 6: The retrospective nature of the study is another critical point as well as the extensive period of recruitment. Gene expression data need to be investigated for a sample-age related bias.

Response: Thank you for this helpful remark. As indicated in the manuscript (line 376-379), the specimen have been resected between 1990 and 2014. As suggested, to address the possibility of an age bias, we have split all cases into two equally sized groups according to the date of surgery. Now, a first PCA plot (Supp. Fig. 1) demonstrates that samples from both categories diffusely overlap, indicating that different sample age did not lead to a shift of the overall gene expression (lines 112-118). In general, the Nanostring system is a very robust method to quantify RNA from FFPE tissue (Technote: https://www.nanostring.com/scientific-content/technology-overview/ncounter-technology/ffpe-simplified). Apart from all target genes, also endogenous housekeeping genes, as well as in-build positive and negative controls are measured separately for each sample. Using these genes, we normalized the expression counts for all samples according to the manufacturer’s instructions (nSolver 4.0 software). This compensates a possible loss of RNA through age-related degradation and practically excludes a technical inter-sample bias. Moreover, all samples met the required amount of input RNA as specified by the manufacturer (lines 404-414).

Major comment 7: Clinical data lack follow-up. The study is merely descriptive in nature and provides no clinical correlate. Expanding the study into a multi-centre setting might enable meaningful interpretation of TME findings in a clinical context.
Major comment 8: Validation in an independent data-set should be performed. If possible, as part of the suggested multi-institutional approach.

Response to major comment 7 and 8: A very good suggestion. We performed immunohistochemistry and RNA-in-situ hybridization on an independent cohort of 15 cases for three significant genes (lines 177-180). Moreover, using mass spectrometry imaging, we validated our results on COL27A1 with a quantitative method (lines 209-231). The lack of clinical follow-up is critically remarked within the limitations (lines 350-353). Unfortunately, it is impossible for us to implement the suggested extension of the study within the outlined timeframe. As indicated above, such an approach would at least take another year to carry out. With a larger cohort, we will address the clinical implications of expression patterns in future studies. Also, we are currently establishing a network with other centers in order to have a validation cohort at hand for subsequent studies.

Major comment 9: Quantitative PCR validation should be considered for all significant genes.

Response: Thank you for this helpful remark. As outlined above, we used three different methods to validate our Nanostring results. Among them, we used RNA-ISH to validate the results on RNA level (lines 174-208). Unfortunately, it is impossible for us to implement the suggested qPCR experiments within the outlined timeframe.

Major comment 10: Histopathological work-up of the study group in insufficiently depicted. Especially molecular studies are not reported (e.g. MYB-NFIB/MYBL1-NFIB/MAML2 associated translocations or fusions).

Response: In order to provide a well-characterized cohort with reliable diagnoses, all cases underwent comprehensive histopathological and immunohistochemical workup, consisting of antibodies against CK7, p63, S100, androgen receptor, HER2, NOR-1. If any uncertainty concerning the diagnosis remained, FISH analyses for MYB/MYBL1 and MAML2 translocations were performed (break-apart probes). Cases with ambiguous diagnosis after the work-up were strictly excluded from the study. A description of this procedure has been implemented in the manuscript (lines 365-376).

Minor comment 1: Please provide an explanation for every abbreviation upon first use (CAF).

Response: The term “CAF” is now explained in the introduction (lines 50, 51). Also, we checked all other abbreviations.

Minor comment 2: Boxplots are unsuitable for the depiction of datasets as small as this one (e.g. n = 9). Individual datapoint should be plotted instead.

Response: Thank you for this valuable suggestion. We revised figure 2, supplementary figure 3 and supplementary figure 4. As you recommended, we added the individual datapoints to increase data transparency, but we also kept the boxplots to allow the perception of gene expression patterns. We hope that you agree, that this design now depicts our results in an appropriate manner.

Reviewer 3 Report

The article is well crafted and treats an interesting and poorly investigated topic, Salivary Gland Carcinomas. The study of ECM may have immediate consequences for the therapy of such neoplasms. I advise for pubblication of this article, but some corrections are needed in my opinion. 

  • First of all, authors should clarify the choice of those particular 28 genes in the different sections;
  • line 33: CAF should be defined prior to use the acronym;
  • ln 346: erase "few"; among the limitations, authors should underline that it is not case-controlled, as you did not assay values on healthy subjects;
  • Methods: authors should clarify how patients were selected; is this a retrospective study in which consecutive patients were included? Specify, underlying inclusion/exclusion criteria if any. 
  • ln 361: specify if you used the latest TNM version; did you reviewed older cases according to latest TNM? If not, it should be done
  • ln 361-362: "For tumor and patient’s characteristics compare table 1": delete this sentence, as it should not be in a "M&M" section being a result; 
  • Finally, English needs extensive revision, as it is insufficient in many passages, and, given the limitations of the study (sample, absence of controls and number of investigated genese above all), the title should underline the explorative/pilot spirit of the study. 

Kind regards

Author Response

Comment 1: First of all, authors should clarify the choice of those particular 28 genes in the different sections;

Response: Thank you for this remark. After extensive literature research on the topic, we extracted all collagens, laminins and other ECM-related glycoproteins from the panCancer gene set. Together with a complete list of all analyzed genes (lines 423-426), this is now clarified in the results (lines 103-104) as well as in the MM section(lines 421-423). We also remarked this among the limitations of the study (lines 346-348).

Comment 2: line 33: CAF should be defined prior to use the acronym;

Response: The term “CAF” is now explained in the introduction (lines 50, 51). Also, we checked all other abbreviations.

Comment 3: ln 346: erase "few"; among the limitations, authors should underline that it is not case-controlled, as you did not assay values on healthy subjects;

Response: Thank you very much. From each patient, we analyzed both tumor and normal tissue. Nonetheless, as you remarked, we cannot exclude that underlying mechanisms such as germline mutations might have affected not only the tumor but also the normal tissue. This is clearly stated in the end of the limitations section (353-356). Also, we deleted the word “few” as you suggested.

Comment 4: Methods: authors should clarify how patients were selected; is this a retrospective study in which consecutive patients were included? Specify, underlying inclusion/exclusion criteria if any.

Response: A very helpful remark. We now declare the retrospective nature of the study (lines 80, 103, 346). In the M&M section, we now clearly state how the cases were selected. In brief, the archives of the Institute of pathology were searched for AdCy, MuEp and SaDu resection specimen. Then, if enough FFPE material was available and if the patients had given their informed consent, the cases were reviewed by two pathologists using a comprehensive antibody panel and break apart FISH assays for MAML2, MYB and ETV6 gene translocations if needed to establish an unequivocal diagnosis. Ambiguous cases were strictly excluded from the study (lines 365-375)

Comment 5:ln 361: specify if you used the latest TNM version; did you reviewed older cases according to latest TNM? If not, it should be done.

Response: Thank you for this comment. All TNM classification were corrected according to the latest

WHO classification of head and neck tumors (8th edition, 2020) (lines 375-376).

Comment 6: ln 361-362: "For tumor and patient’s characteristics compare table 1": delete this sentence, as it should not be in a "M&M" section being a result;

Response: Thank you for this remark. We deleted this sentence as you suggested.

Comment 7: Finally, English needs extensive revision, as it is insufficient in many passages, and, given the limitations of the study (sample, absence of controls and number of investigated genese above all), the title should underline the explorative/pilot spirit of the study.

Response: We critically reviewed wording and grammar. Additionally, we had a professional lectorate review the entire manuscript. We hope that this improved the linguistic quality to a large extend. Also, we changed the title of the manuscript to “A panel of 28 extracellular matrix genes reveals global and entity-specific characteristics in adenoid cystic, mucoepidermoid and salivary duct carcinomas”, indicating that we analyzed neither all ECM-related genes nor all SGC subtypes.

Round 2

Reviewer 1 Report

This is a unique discovery study looking into extracellular matrix (ECM) of salivary gland cancers (SGCs). The authors used the Nanostring assay to explore RNA expression of ECM related genes in 3 different SGCs. Inherently the study is limited by its sample size - there were total of 34 patients with 11 adenoid cystic ca (AdCy), 14 mucoepidermoid ca (MuEp), and 9 salivary duct ca (SaDu). It is interesting to note that hierachial clustering differentiates AdCy from the others, which were later confirmed with RNA ISH, IHC and mass spec.

  1. I do think this is a very original study, but I would suggest to elaborate more on how these information would have translational potential.
  2. Section 2.6 on immune infiltration seems to be un-related to the topic of the manuscript. Also, this section contains inaccurately labeled all CD8+ T cells as 'cytotoxic' T cells, while regulatory T cells are known to be CD8+. Likewise, CD68 can be expressed in other cells such as neutrophils, and cannot reliably be used as a definite marker for macrophages. I would suggest to take this section out completely. 

Author Response

Thank you very much for your very constructive feedback.

  1. This is a helpful suggestion. We have implemented a short paragraph which highlights the fact that several significantly up-regulated ECM components are currently under intense functional or clinical investigation as therapeutic targets. Thus, our findings are of high translational relevance and provide a strong rationale to dissect these molecules' functional role also in SGC (lines 300, 336-343.
  2. These complementary analyses were added after a suggestion by one of the other reviewers. In fact, we support your position that this additional topic distracts the manuscript's focus rather than adding novel insights as we and other authors have already described the immune infiltrate of SGC. Consequently, we have taken out this entire part from the current manuscript. We have elaborated our opinion on this subject in a letter to the editor-in-chief who will determine if this section will be included in the manuscript or not.

Reviewer 2 Report

The authors have responded to all of my concerns in a reasonable fashion. Major issues were resolved and minor flaws are now openly discussed.

The readability was greatly enhanced. As the manuscript is now more informative yet transparent regarding its methodical shortcomings.

I'm impressed with the author's rapid and rigorous revision of the manuscript. Thanks for the appreciative feedback especially in the light of my rather skeptical first review.

I suggest the manuscript be accepted in its current form.

Author Response

Thank you for your constructive feedback which has contributed to improve the final manuscript.